# Functional Weight of Somatic and Cognitive Networks and Asymmetry of Compensatory Mechanisms: Collaboration or Divergency among Hemispheres after Cerebrovascular Accident?

**DOI:** 10.3390/life11060495

**Published:** 2021-05-28

**Authors:** Hélène Viruega, Manuel Gaviria

**Affiliations:** 1Institut Equiphoria, Combo Besso-Rouges Parets, 48500 La Canourgue, France; helene.viruega@equiphoria.com; 2Alliance Equiphoria, 4, Résidence Le Sabot, 48500 La Canourgue, France

**Keywords:** cerebrovascular accident, adaptive neuroplasticity, maladaptive neuroplasticity, brain asymmetry, within-network compensation, across-network compensation, intercallosal processes, brain homeostasis, neurorehabilitation

## Abstract

The human brain holds highly sophisticated compensatory mechanisms relying on neuroplasticity. Neuronal degeneracy, redundancy, and brain network organization make the human nervous system more robust and evolvable to continuously guarantee an optimal environmental-related homeostasis. Nevertheless, after injury, restitution processes appear dissimilar, depending on the pathology. Following a cerebrovascular accident, asymmetry, within- and across-network compensation and interhemispheric inhibition are key features to functional recovery. In moderate-to-severe stroke, neurological outcome is often poor, and little is known about the paths that enable either an efficient collaboration among hemispheres or, on the contrary, an antagonism of adaptative responses. In this review, we aim to decipher key issues of ipsilesional and contralesional hemispheric functioning allowing the foundations of effective neurorehabilitation strategies.

## 1. Introduction

Human brain can be defined as a highly multifaceted, interconnected network of specialized cell populations that balances functional regional segregation and specialization through robust integration. This balance generates complex and accurate coordinated dynamics across multiple spatiotemporal scales [1,2,3]. In order to guarantee an efficient behavior to individually adapt towards a permanent changing ecosystem, the brain is able to run reprogramming actions allowing either, a rapid update of motor plans, or a withdrawal of a prepared action with a shift towards an alternative action [4].

However, how changes in brain network structure and dynamics relate to behavior is fairly unknown. Reducing characteristics of those networks to a single value renders a poor and often misleading guide for understanding brain’s possibilities [5]. A deeper knowledge of the human brain adaptive mechanisms and paths can, at best, be achieved through mathematical modeling of these complex phenomena provided consideration of the many inputs and systems simultaneously activated when performing a real-life task. However, even though these complexity-driven methodologies offer powerful ways to decipher the dynamic interactions of brain regions to produce emergent behaviors in health and disease, the impact of lesion on complex network is largely unknown [6].

One of the brain’s fundamental properties is the ability to adapt its processes to a wide range of complex external/internal modifications, including those generated by injury. This inherent neural plasticity promotes efficient functional outputs in healthy individuals or, at least, some degree of functional outcome in disease. In the latter, spontaneous mechanisms of repair are rarely sufficient to support significant long-term recovery [7]. Three major characteristics are crucial for the implementation of efficient neuroplasticity after injury: compensation, degeneracy, and reserve. It is expected that patients with higher degeneracy and reserve will be better armed to adapt to the functional impairments following a brain injury [8].

Cerebrovascular accident (CVA) is a devastating event and a leading worldwide cause of mortality and disability. According to the WHO [9], nearly 17 M individuals suffer a CVA each year, which add to a pool of 33 M stroke survivors. As illustrated by the number of people that remain disabled after a CVA (2 out of 3), the extent of recovery is limited, and novel therapeutic approaches are urgently needed. Many hopes and expectations have been recently placed on neuro-pharmacotherapeutics through the use of high-throughput screening and computer-aided drug design for an optimal identification and validation of molecular targets. However, even though potentially complementary to neurorehabilitation, neuropharmacology must deal with substantial barriers. Indeed, drug discovery and development are seriously hindered by the incomplete understanding of the pathophysiology of neurological diseases and most of the pharmacological treatments target the symptoms instead of the cause of the disease [10,11]. In this context, given the highly redundant configuration of the brain, compensatory mechanisms of neural networks potentially allow a variable degree of functional recovery. Experimental and theoretical model predictions have until now enriched our knowledge about brain’s functioning after an injury. However, the archetypal narrow experimental action-reaction approach (i.e., mechanistic point of view) constitutes a huge barrier for the understanding of some crucial issues. Those strongly determine the diversity of pathology-dependent and individual-dependent neurological outcomes and might be profoundly questioned.

CVA neurorehabilitation might include a rational functional and structural approach relying mainly on the connectome and its dynamics. In this context, affected, at-risk, and preserved networks should be identified and targeted with specific single and time-dependent strategies after injury. These strategies must take into consideration the hemisphere asymmetry, the interhemispheric functional inhibition phenomena, the intrahemispheric and interhemispheric plastic phenomena, and the alterations at a distance from focal lesions [12]. Moreover, the functional rehabilitation approach should include how perception, emotion, motivation, and cognition (attention, executive control, working memory, planning …) interact and lead to an improved outcome, and how the activity of brain regions involved in such diverse domains is coordinated. Under physiological conditions, emotional and motivational processing shapes brain responses by increasing functional connections across dissimilar brain hubs and regions. It has been hypothesized that such interactions lead to improved behavioral performance during challenging tasks, and that networks can be considered as dynamic processes whose evolution is closely tied to the underlying mechanisms supporting behavior [7]. Brain function, dysfunction, and reshuffle after CVA must be put into perspective well beyond the conventional boundaries differentiating action, cognition, and emotion [13,14], taking into account the “priorities” of the patient when functional recovery is initiated.

## 2. Brain Asymmetry and Hemispheric Specialization: Increasing the Speed and Efficacy of Information Processing

Brain hemispheres exhibit anatomical and molecular left-right asymmetries correlated with functional specialization. This unique conformation allows the individual to optimally perceive and respond to environmental stimuli. Asymmetry and functional specialization have been extensively described with respect to cognitive and emotional functions. For instance, asymmetry has been observed in a broad range of functional processes and attributes. Left dominance has been demonstrated for episodic memory encoding, positive emotional valence (i.e., approach), risk-taking, language, problem solving, logical interpretations, and viewing details. Right dominance has been reported in episodic memory retrieval, pseudo-neglect, negative emotional valence (i.e., avoidance), impulsivity, face processing, global viewing, and visuospatial tasks [15,16]. Throughout evolution, lateralization has been assumed to provide functional advantages [17]. Indeed, it impedes processing duplication to maximize brain tissue usage and enables the hemispheres to perform multiple simultaneous tasks through partitioning [18] and to effectively process information in the shortest time by preventing trans-callosal transmission [15]. Besides, it has been suggested that aging is accompanied by a decrease in asymmetry which reflects compensatory plasticity and would be associated with better cognitive performance due to the participation of the non-dominant hemisphere.

Even though hemispheric specialization has been extensively acknowledged in several cognitive domains, a question remains as to whether the localization of one function to a hemisphere predicts the localization of another correlated function to the opposite hemisphere. This has been considered as a complementarity of brain asymmetries among hemispheres. However, the different functions of the hemispheres have been currently assessed individually, with little attention to the relationship in the degree and direction of lateralization [19]. This is a key issue since complementarity arises not only in a single cognitive domain but also across cognitive domains (i.e., it is difficult to imagine the activation of the short-term memory without the activation of the attentional function, etc.) and might operate alike in other spheres (namely the somatic or emotional domains) in a ripple effect. Given the wide array of asymmetrically organized human brain functions and outputs, it is crucial to understand how and why this asymmetry is implemented and entwined [20].

Since both hemispheres contribute to multidimensional cognitive, emotional, and somatic domains, complete hemispheric specialization is controversial. Indeed, both hemispheres have capabilities to execute a same task with different degree of refinement [21]. But are they able to fully take over a function when the other hemisphere is failing? In other words, could the intact hemisphere take in charge both the disrupted contralateral complex functional outputs and the homolateral complementary preserved ones? Does asymmetry have a functional threshold beyond which functional restitution is unsuitable for the healthy side?

Data on stroke behavioral outcome are consistent with several features of asymmetrical plasticity. Those include the transient role of contralateral hemisphere in recovery of lateralized functions, the disparate recovery levels of highly lateralized functions such as language or motor function, and the reduced recovery of functions that are not typically highly lateralized [15]. However, it is not clear whether in this case the hindered compensation from the non-lesioned hemisphere is the consequence of the intrinsic characteristics of the involved circuits (namely the number of intra-and inter-hemispheric connections, i.e., functional asymmetry) or concerns the asymmetry of receptors, dendritic spines, and molecular machinery involved in a specific functional output (i.e., structural asymmetry).

## 3. Functional Optimization of Neural Networks: “Small-World” Topology Allows Maximal Communication Speed with Minimal Energy

The adult human brain is less than 2% of the body’s volume (~1.4 kg for 70 kg) but burns ~23% of daily caloric intake (i.e., ~415 kcal/day). In the newborn (~0.4 kg for 3.5 kg), this ratio is even more disproportionated since the brain burns around 75% of total daily caloric intake (i.e., ~120 kcal/day). Each neuron consumes approximately 4.8 × 10^−6^ cal/day: it may fire around 350,000 times/day at a rate ranging from 0.15 to 16 Hz and each action potential may consume around 1.19 × 10^8^ ATP. One action potential of a cortical neuron per second raises oxygen consumption by 145 mL/100 g gray matter/h [22]. The spontaneous brain activity accounts for 70% of the energy consumed by the brain and thus, at a whole brain level, basal metabolism is estimated to consume 30% of brain glucose. Thus, the high energetic cost of the human brain function can only be held through a combination of strategies for efficient energy use [23].

An outstanding level of refinement is pivotal to yield developmental complexity, cellular and molecular uniqueness, activity-dependent plasticity, signal processing ability, and network organization, which are altogether brain’s unique features [24]. The brain is typically organized to produce high value for low cost. Its topological organization throughout neural networks is crucial for its overall function, performance, and behavior. Indeed, since it is a costly system to build and to run, brain networks are shaped to have high topological efficiency, robustness, and modularity [25]. Accordingly, the brain shows high global efficiency of information transfer between brain regions located far away from each other in the anatomical space [26].

Many aspects of brain organization presumably depend on the principle of minimizing the wiring cost involved in anatomically connecting neurons to form networks. In general, it can be assumed that the cost of building and maintaining axonal connections, as well as the speed of signal transmission, increases with the volume of wiring and the distance of the neuron-to-neuron connections. The increase in the number of neural elements in the human brain has required an increase in the number of connections, hence imposing additional wiring costs.

In addition to the costs of building an anatomical brain network, the costs of running it must be considered. While the metabolic costs of the brain are disproportionate in relation to the body (i.e., 10 times higher than what would be expected from its weight alone), they are thoroughly controlled to be as low as possible for any given function [27,28]. Nevertheless, brain’s volume expansion has an uneven impact on the whole metabolic cost of the body. Active maintenance of electrochemical gradients across neuronal membranes accounts for most of the substantial brain’s metabolic cost [27]; these are pulled down by myelination and pulled up by axonal length and diameter, long-distance connections being metabolically more expensive to maintain [29]. Besides, by minimizing the length of anatomical connections in the network (i.e., pulling down the wiring costs), the system will also regulate running-dependent costs.

The organization of neural networks follows two topological properties; it is both random and regular: it is random in the sense that the neural path-lengths tends to be short, and regular since the network configuration shows high degree of clustering. Thus, neural networks typically have high aggregation properties and high overall efficiency [26]. In this specific context, a trade-off between efficiency and connection distance can be quickly renegotiated by the brain depending on the complexity of the task. For example, when there is a great demand for cognitive processing, neural networks adopt a more efficient but more costly workspace configuration. On the other hand, when cognitive demand is lower, the brain networks move to a more clustered and less costly configuration [30].

Under normal physiologic conditions, interactions between neurons and glial cells are vital to meet the energy needs of the brain but are also important in the control of many essential brain functions such as homeostasis of the body and memory consolida-tion. Neuronal metabolic processes directly depend on the activity of astrocytes, which produce lactate and activate glycolysis and glycogen metabolism [31]. Indeed, astrocytes store glucose as glycogen, which can be temporarily used for oxidative metabolism, leading to the generation of lactate, which can, in turn, be shuttled to neurons as an energy source [32,33,34,35]. All these metabolic issues gave rise to the astrocyte-neuron lactate shuttle (ANLS) hypothesis, whereby glutamate released in the synapsis and its reuptake into astrocytes triggers glucose uptake into the brain parenchyma and lactate production by astrocytes for the use of neurons [34,35]. The transfer of lactate from astrocytes to neurons is one example of the wide palette of metabolic relationships between these cells. Lactate in the brain has long been associated with ischemia but it is now considered a main regulator of the brain’s ‘homeostatic tone’, by ensuring adequate energy supply, setting neuronal excitability levels, and regulating adaptive functions that are mediated by plasticity mechanisms (e.g., memory). The ANLS model has also been extended to metabolic exchanges between oligodendrocytes and axons showing that, in animal models, lactate released by oligodendrocytes is required to maintain axonal function [34].

The brain is highly vulnerable to any condition that threatens its energy supply. After injury, the main degenerative mechanisms converge to loss of intracellular homeostasis with massive energy failure and cell death. Neuronal survival is multifaceted and encompasses well-fueled energy metabolism, trophic input, clearance of toxic substances, appropriate redox environment, integrity of blood–brain barrier, regulation of programmed cell death pathways, and cell cycle arrest. Mechanisms of delayed degeneration entangle excitotoxicity subsequent to hyperexcitation of glutamatergic receptors, loss of intracellular calcium homeostasis, energy failure, endoplasmic reticulum stress, reactive oxygen species production, neuroinflammation and axonal degeneration of both synaptic inputs (anterograde degeneration), and projection targets (retrograde degeneration) [36].

Some intrinsic mechanisms of neuroprotection are triggered by the nervous system after an ischemic injury. For example, according to some studies in animal models of cerebrovascular ischemia, glycogen increases in the penumbra within three days post-injury and colocalizes with astrocytes [37]. The rise of glycogen might temporarily neuroprotect at-risk cells through increasing the available metabolic substrate. These molecular mechanisms of energy supply driven principally by astrocytes, although mostly insufficient for the overall tissue viability and function preservation, are mandatory for supporting intra-hemispheric neuroplasticity. They protect the cells of the penumbra that are essential contributors for behavioral restitution.

Understanding neuronal death with regard to the overall cellular and molecular mediators inside an integrated dynamic system is a tricky issue just as the understanding of the reasons for failure of the regeneration path in humans is [38]. The rationale for implementing neuroprotective and/or neuroregenerative therapeutic strategies after CVA is however beyond the scope of the current review.

Several questions arise since the fundamental insight of Ramon y Cajal in 1899 [39]: “All of the various conformations of the neuron and its various components are simply morphological adaptations governed by laws of conservation for time, space and material”. Those ‘laws of conservation’ are probably accountable for the final functional outcome which relies on neuroplasticity. For instance, if the reconfiguration costs compromise the whole-brain homeostasis, the behavioral restitution will be seriously endangered. Energetically speaking, the question is: Does the redundant architecture of the brain safely enable a rearrangement of its individual fundamental units regardless of the time elapsed since injury?

## 4. Neuroplasticity Mechanisms: Within and Across-Network Compensation

Little is known on how the intact healthy human brain reshapes itself in response to a focal disturbance in the short- and long-term. From a network perspective, to compensate a deficit of a key function, the interaction leading to adaptative plasticity within and between neural networks remains fairly unclear [40].

Besides cortical regions that are specialized for domain-specific processes (e.g., motor perception, early visual and auditory processing, …), the brain also comprises cortical regions driving more general cues. These so-called domain-general brain systems are engaged across a wide range of cognitive and emotional tasks. Two sets of large domain-general cortical regions can be mentioned: the “multiple-demand network” and the “default mode network”.

The “multiple-demand network” would involve general functions such as maintenance and direction of attention, cognitive control and flexibility, error monitoring, behavioral inhibition, or short-term working memory processes that are necessary for all goal-directed motor and cognitive tasks. Those regions would have the capability of rapidly adapt through a top-down control drive during a broad range of tasks. This system would be minimally engaged when the individual performs an overlearned task such a routine task. Conversely, it would be triggered when addressing novel problems, when a task condition changes and usual response requires adjustment, and in a broader sense, when a high level of top-down task’s control is required [41]. On the other hand, the “default mode network” includes brain regions that show an increased activity during task-unconstrained rest periods and deactivation during specific cognitive tasks. This network has been associated with some specific tasks including episodic memory, prospection into the future, social cognition, spatial navigation, perspective-taking, and semantic processing [42,43]. The increased activation of these two domain-general systems would reflect recruitment of general mental resources when domain-specific task demands substantially rise.

A growing body of knowledge stresses the fact that some brain mechanisms may support compensatory flexibility of neural networks during or after a perturbation. The first, second, and third mechanisms concern within-network compensation whereas the fourth and fifth refer to across-network compensation:-The first one relies on the resilience and robustness of the given network that potentially enables compensation and allows the region to maintain an adequate level of functional processing. In this case, the activity pattern itself might not change after the perturbation but its level of specific contribution would increase [44,45,46].-The second one is based on the activation of other neighboring nodes within the same network to supply the function by bringing their network contribution to the performed task [47].-The third one allows the upregulation of contralateral homologous regions, not usually contributing in a strong manner to the current task processing, to help maintaining functional activity [48,49,50].-The fourth mechanism engages the recruitment of alternative pathways relying on neuronal degeneracy (i.e., the capability of other neuronal systems not belonging to the current neural network to carry out the same function than those disrupted) [5,51,52].-Finally, the fifth mechanism consists of the recruitment of domain-general networks that also provide compensatory behavioral restitution if domain-specific networks are disturbed [53].

These mechanisms could support functional adaptability and robustness against focal brain lesions and, remarkably, would not be mutually exclusive.

Evidence for rapid within-network compensation has been provided in healthy individuals after transcranial magnetic stimulation (TMS). The elicited perturbation generates the inhibition of the targeted region and of remote regions of the same network. In this case, the upregulation of contralateral homologous regions and ipsilateral network nodes allows to sustain task processing following the virtual perturbation. This mechanism has been demonstrated across several domains including attention, action planning, working memory, auditory cognition and language elaboration and production [4,48,49,50,54,55,56,57].

Short-term reshaping evidence also involves regions outside the current network providing some proof for across-network compensation. For example, in the case of the language network, a perturbation of a phonological region triggers the activation of a neighboring network through some degree of degeneracy showing the compensatory potential of other networks. This case illustrates the notion that a perturbation may yield to a prompt adaptative response within the non-targeted network to compensate the decrease of task-specific neuronal activity [58].

A high degree of flexibility of neural networks in terms of distributed processing has been globally characterized and likely relies on the potential contribution and rapid reconfiguration capabilities of a single region within a network and across networks. The recruitment of within- and across-network regions seems to depend on the degree of disruption of the specific function. However, some of these conclusions about the dynamic regulation of intrahemispheric interactions have been obtained by studying the healthy human brain and the functional disruption has been elicited by virtual suppression of task-specific neuronal activity. The question is whether these results are totally relevant and can be entirely extrapolated in pathological conditions such as brain stroke where a myriad of functions are disrupted simultaneously and are intertwined in a complex manner.

## 5. Interhemispheric Dialog: Opening and Closing Gates

The corpus callosum is the main structural path tying contralateral brain regions. In humans, it has around 200 million mostly myelinated fibers and is required to integrate bilateral motor and sensory signals enhancing somatosensory input detection and discrimination [59,60]. It is a key structure supporting the emergence and preservation of brain asymmetries, showing itself some degree of structural asymmetry that will also affect interhemispheric transfer [61,62]. Most of the callosal axons are excitatory glutamatergic axons but they target inhibitory interneurons potentially inhibiting the contralateral hemisphere [63].

The emergence of hemispheric asymmetries in the human brain through the corpus callosum most likely relies on both excitatory and inhibitory models [64]. According to the excitatory model, functional hemispheric asymmetries occur following conduction delay during interhemispheric information transfer. A longer conduction delay would lead to the performance by one hemisphere of time-sensitive processes to ensure faster outputs, significantly reinforcing asymmetry [65]. In contrast to excitatory theories, inhibitory models assume that the homotopic areas of the hemispheres would mutually inhibit each other through the corpus callosum aiming to highly refined degrees of behavioral control [66]. In this case, adjacent areas within one hemisphere would mutually inhibit one another which leads to the following: (i) the surrounding area in that hemisphere would be activated; and (ii) the activated area would inhibit its homotopic area in the other hemisphere. Both functional models of the corpus callosum are conceivable and the precise functional correlation would depend on the callosal subsegment, the involved fibers, the targeted interneurons, and the brain regions in the contralateral hemisphere [63]. As a result, both hemispheres could simultaneously become dominant for different performed processes in closely neighboring areas of one hemisphere for complementary functions (i.e., become dominant for two different ways of processing information in the same general homotopic brain regions).

After unilateral perturbations such as stroke, interhemispheric connectivity is altered, and it is supposed to often lead to early bilateral somatomotor cortical hyperactivity (in patients with modest recovery). In this context of unilateral disruption, there can be substantial changes in bilateral cortical responses, either beneficial or detrimental for recovery. However, the molecular and cellular changes, including synaptic phenomena, underlying these large-scale structural and functional interhemispheric alterations are only little understood [67].

An increased amplitude of cortical somatosensory potentials contralateral to a lesion has been frequently observed and is explained by the loss of inhibition, normally exerted by the injured cortex. This loss of inhibition would be mainly due to the interruption of the transcallosal pathway mechanisms [6]. However, contralesional activation seems to be size-dependent (i.e., bigger for large lesions). From a functional point of view, it appears to be helpful for large lesions and harmful for small ones. In the latter, an initial contralateral activation is quickly interrupted and replaced by the activation of perilesional areas of the injured hemisphere [68]. These effects might be controlled at the level of the callosal pathways, but little is known about their exact role in interhemispheric plasticity [67]. A concrete example is the relationship between the two primary motor cortices. Activity of the ipsilateral motor cortex is controlled in some measure by the contralateral one. Thus, in healthy individuals, the inhibitory effect of one motor cortex on the other decreases depending on the movement characteristics when executing a motor task [69]. Persistence of a bilateral activation pattern in stroke patients could be associated with insufficient recovery [70].

Understanding the interhemispheric dialog and its dynamic rules might enhance our capacity to strengthen patient recovery from stroke by designing more specific, targeted, and timely therapeutic interventions.

## 6. Stroke and Adaptative Neuroplasticity: A Complex “Pas-de-Deux”

In the last two decades, new evidence has been provided with respect to the mor-pho-functional characteristics of brain circuits and the role of glial cells in all aspects of brain function including neural plasticity. Radial glia, astrocytes, OPCs, oligodendrocytes, and microglia each influence nervous system development (i.e., neuronal differentiation, migration, axon specification and growth, circuit assembly, and synaptogenesis). With neural circuit maturation, each glial type fulfils key roles in synaptic communication, plasticity, homeostasis, and network-level activity [71]. Interestingly enough, astrocytes are intimately associated in the so-called “tripartite synapse”, receiving signals from the presynaptic neuron and responding by releasing feedback signals. These glial cells are integral modulatory components of the synapse. Besides fast neurotransmission, astrocyte regulation of synaptic transmission can run on a different time scale. Thus, astrocytes can transitorily control the synaptic strength, contributing to long-term synaptic plasticity. They are cellular processors of synaptic information and can regulate synaptic transmission (and plasticity) by treatment, transfer, and storage of information. Concomitantly to neuronal populations, they must be considered as key elements involved in brain function and potential targets for functional refinement in health and disease [72,73].

As described in the above sections, even though various neuronal mechanisms for the optimization of neural plasticity are strong and closely related (i.e., degeneracy, compensation, and reserve), they do not necessarily mean that compensatory activity of intact regions will occur following an injury. Indeed, atypical activation will occur when for example, degeneracy is partial or incomplete [6].

In addition, mechanisms for compensation may be maladaptive and worsen the functional outcome. Three main maladaptive responses (two functional and one structural) are incriminated in the spreading of the functional deficit throughout the connectome: diaschisis, transneuronal degeneration, and dedifferentiation [5]. Of these, diaschisis is a temporal interruption of functional connectivity away from the injured site. In this non-structural alteration, the distant neurophysiological changes must correlate with behavioral changes and those changes must move towards normalization with time [6]. Alterations of functional connectivity may appear in regions that are not directly linked to the lesion. Interestingly enough, diaschisis may be context-dependent, which means that it may be apparent only when performing some tasks [74]. It may fluctuate as a function of the type of injury and depends on how the injury impacts the overall neural dynamics instead of its local effect on the lesioned site alone [5,75]. Furthermore, the ability to activate functionally altered regions at a distance depends on whether the stimulus comes from the damaged zone or from another brain region [6]. In this specific context, merely tracking the spread of an injury will not automatically explain the mechanisms that are accountable for its dissemination. In the long term, diaschisis may precede more structural changes such as transneuronal degeneration characterized by neuronal shrinkage, reductions in dendrite and synapse number, alterations of axonal myelin content, reduction of fiber number, and neuronal death [5]. The decrease of excitatory input and the loss of trophic support from damaged areas should reinforce those structural alterations [76].

Dedifferentiation is the second main non-structural maladaptive response after injury. It consists of the diffuse, unspecific recruitment of brain regions to accomplish a task. It appears to be the consequence of a failure of typically specialized neural activity and a disruption of balance between excitation and inhibition of neural systems [5,77,78].

However, the brain is also capable to respond to an injury throughout adaptive mechanisms that enable, when possible, to sustain homeostasis, signal processing, and functional output [79]. In shaping a post-injury brain, experience continuously interacts with genetic information and “obtainable” molecular, cellular, and functional resources. The final functional phenotype is the combined work of the individual available biological substrates and the individual experience. Synaptic plasticity involves both functional and structural changes. Neuromodulators such as norepinephrine, acetylcholine, dopamine, or serotonin, which reflect the level of arousal, motivation, attention, affection, and emotion of an individual, are powerfully involved in the initiation and maintenance of such plasticity [80]. In this context, structural plasticity is a crucial neural substrate for compensation. Indeed, after a vascular injury leading to focal ischemic damage, a wide depolarization of connected regions can follow. This will produce a persistent hyperexcitability and/or a disinhibition of functionally related networks accompanied by sprouting of undamaged axons and a certain degree of synaptogenesis [81,82,83]. These structural changes can occur remote from the focal damaged site and would offer enhanced flexibility to support functional plasticity aiming to preserve, as far as possible, the function [82].

Cortical reshaping is concomitant to robust and durable changes in structural connectivity relying on axonal sprouting and dendritic spine turnover. However, there is no certainty that these new structural pathways are indispensable for the regaining of function. It might, on the contrary, represent maladaptive transformations that contribute to the functional shortages [12].

## 7. Neurorehabilitation: Finding the Right Rhythm

Neurological disorders including cerebrovascular accidents often generate severe long-term disabilities with substantial day-to-day consequences. Deeper understanding of brain functioning in the last two decades has opened new perspectives for more integrative interventions, boosting the intrinsic abilities of the brain for functional compensation. Neurorehabilitation capitalizes on neuroplasticity that occurs throughout life. A myriad of theories regarding brain functioning after injury have been put forth but current clinical management of severe disabilities after CVA is often ineffective or not totally conclusive (e.g., the use of therapeutic hypothermia has been proposed as promising, but results are partly questionable mainly due to its efficacy in hemorrhagic stroke, its side effects, and the invasive nature of the procedure [84]). When beneficial, the mechanisms of action leading to the favorable outcome are to a large extent unclear [7]. Indeed, identification of optimal treatment approaches to improve functional outcome is limited by the incomplete understanding of the neurobiological principles of recovery.

Certain clinical findings cannot be explained by the theoretical consequences of the disruption of a particular and well-characterized functional area of the brain. We must consider that the loss of a neurological function after an injury can result from abnormal plastic arrangements that preferentially disrupt the most energy-consuming components of the concerned neural network. These high energy-consuming mechanisms are, most of the time, vital for the integrative treatment and the adaptive behavior of the system [25,26], which creates a neuroplastic paradox.

A second drawback arises from the laws that support functional adaptability against lesion. Indeed, the non-mutual exclusivity of brain mechanisms for compensation adds a second level of complexity where synergies for functional restitution are replaced by energy-dependent competition. Thus, the available brain mechanisms for compensation will respond to the less-expensive running costs, which is not necessarily the most effective pathway.

Homeostasis is a major issue for every post-injury situation, no matter how far away the injury has occurred. It is well known that the brain plays a fundamental role in the regulation of energy homeostasis. The brain coordinates a fine-tuned control both on behavioral patterns and peripheral metabolism throughout the assessment of diverse signals (for example, neuronal, endocrine, metabolic) that mirror the instantaneous energy status and its relation to the potential requirements [85]. Humoral signals convey the metabolic status from different tissues to various brain regions enabling the brain to continuously regulate energy homeostasis. Amongst them, circulating levels of glucose and free fatty acids, are essential metabolic substrates integrated to the signals coming from the periphery [86]. In this context, if the reconfiguration costs compromise the whole-brain homeostasis, the restitution of a process will be delayed or simply switched off, which raises the question of the right timing for the different therapy blocs.

Experimental and theoretical neurobiological models have dramatically widened our knowledge of the brain functions and their vast and sophisticated intrinsic capabilities for managing disruption. At the same time, they have revealed the complexity of the entanglement of the somatic, cognitive, motivational, and emotional spheres. However, so far, no model has been able to comprehensively capture the mutual influence exerted by different processes across the conventional boundaries that segregate action, cognition, and emotion [7]. In this sense, extrapolation from animal models and virtual lesions in healthy individuals are not representative enough [87,88]. In a pathological circumstance like stroke, the comprehension on how connectivity in different functional networks is affected by a focal lesion is highly problematic. Indeed, even though it is possible to determine the impact of a focal lesion on the global brain network architecture, it is less obvious to investigate whether this network impact will have a clinical significance as well as the direction and strength of the relationship between brain regions and the distinction of excitatory and inhibitory polysynaptic effects [6].

Several main phenomena can constitute strong obstacles for behavioral compensation: inter-callosal inhibition, diaschisis, and dedifferentiation. Each of these results in inhibition of adaptive neuroplasticity and potential failing or delay of the ipsilesional recovery mechanisms (i.e., within-network compensation).

When the non-injured brain hemisphere takes control of the disrupted contralateral function, it increases the asymmetry of the function and decreases the interhemispheric complementarities in in-between homotopic areas. In a sense, it takes in charge the main and complementary processes that enable an effective specialized functional output. It is plausible that some neighboring areas of this hemisphere will contribute to restore a certain degree of complementarity through a phenomenon of within-network compensation. Indeed, it is quite unlikely that the same pool of neurons can process simultaneously both, main and complementary processes of a given function. Accordingly, neuroplasticity is activated both to compensate a disrupted process due to the cerebrovascular pathology and to complement the pool of healthy neurons that assume the control of the functional restitution in the contralateral brain through within-network strategies. Those newly activated networks will, in turn, inhibit their homotopic areas in the damaged hemisphere increasing the extent of temporarily functional disruption. Early strong solicitations aiming to restore complex motor skills (postural balance gait, fine motor skills of the upper limb) through massive activation of the cognitive layers (attention, concentration, working memory, sequencing, etc.) should trigger a vast maladaptive neuroplastic response potentiated by negative emotional cues. It is worth remembering that emotional and motivational processes will strongly modulate brain responses by increasing or decreasing functional connections across brain regions modifying performance during challenging tasks [7].

On the contrary, solicitation of pre-elaborated sensitive and sensory processes through multimodal stimulation of an enriched and inspiring environment, excluded a ubiquitous cognitive processing, would easily enable, following across-network early compensation (i.e., bilateral somatomotor cortical and subcortical hyperactivity), a prompt within-network activation of the injured hemisphere attenuating the maladaptive neuroplastic probabilities. Once these mechanisms are triggered, and an initial restitution is launched, the refined cognitive layers can be requested for consolidating the treatment of the processes and integrating the new schemes for progressively improving functional restitution.

Whether this first divergency (functional take over from the healthy hemisphere) can significantly hinder the interhemispheric dialog and consequently the compensation process, the dialog can also be jeopardized by diaschisis. Indeed, alterations of functional connectivity driven by the lesion can be triggered in the non-injured hemisphere. However, since these alterations can be circumvented if the stimulus comes from elsewhere from the damaged zone, it is conceivable that neighboring regions continue to activate the non-injured hemisphere (low running costs) additionally contributing to asymmetry, maladaptive neuroplasticity and divergency. In this case, the preserved skills must be extensively itemized by the therapist and inputs that potentially activate them should be initially restricted until both hemispheres can simultaneously become dominant for different performed processes again (no matter the quality of the restituted process). If the specialized neural activity is thus preserved, the probability of emergence of dedifferentiation should significantly decrease, even if it is expected that some degree of activation occurs in the domain-general brain systems.

Every individual has their own unique connectome. Every individual has their own unique life history and personality construct. Moreover, stroke patients have multiple confounding factors such as age, gender, circadian rhythms, homeostatic mechanisms, and comorbidities, all influencing functional connectivity [12]. How fundamental are these in developing optimal neurorehabilitation strategies remains to be explored. Some points of discussion appear to be crucial: the intrinsic rhythms of the patient’s processes, the right entry point (i.e., the initially targeted sphere by the therapist), the silent barriers linked to insufficient neuroplastic responses (inhibitory pathways), the relevance of therapeutic strategies (as perceived by the patient), their sequencing, and their potential viability with respect to homeostasis considerations. All of them depend on the self-identified priorities of the patient when functional recovery is engaged.

With that in mind, we can legitimately ask: where are the frontiers of neurorehabilitation and what are the boundaries for moving forward? After all, as Carlo Rovelli, the renowned theoretical physicist, so rightly said, “The very foundation of science is to keep the door open to doubt. Precisely because we keep questioning everything, especially our own premises, we are always ready to improve our knowledge”.

## Data Availability

Not applicable.

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
