# Peer review of "Functional Weight of Somatic and Cognitive Networks and Asymmetry of Compensatory Mechanisms: Collaboration or Divergency among Hemispheres after Cerebrovascular Accident?"

_life, 2021, doi:10.3390/life11060495_

Round 1

Reviewer 1 Report

Minor Compulsory Revisions:

This is a reviewed paper on the “Functional weight of somatic and cognitive networks and asymmetry of compensatory mechanisms: collaboration or divergency among hemispheres after cerebrovascular accident?” The authors are experts in this field and have done a nice job in summarizing. The paper is clearly and concisely written. However, there are few suggestions that improve it before publication.

  • Author must provide list of Abbreviations that all are used in manuscript for readers. It is difficult to follow short forms in manuscript if full list of Abbreviations is not available.
  • On page 4, Author state about that fundamental insight of Ramon y Cajal in 1899: All of the various conformations of the neuron and its various components are simply mor-..

          Missing reference of Ramon y Cajal in 1899 in reference list so, please add that reference in final list.

  • On page 5, The first three mechanisms concern within-network compensation whereas the last two refer to across-network compensation:

         Author already give no from first to fifth, so I would like to put it as in sentence that The first three mechanisms concern within-network compensation whereas the last two 4th and 5th one refer to across-network compensation

  • No of subtitles are wrong after no 5. Interhemispheric dialog: opening and closing gates

         Please correct and start that from onwards 6. Stroke and adaptative neuroplasticity: a complex “pas-de-deux”

Author Response

Dear Reviewer,

Thank you very much for your comments that led us to improve the quality of our work.

You can find bellow the answer point-by-point of your remarks:

  1. Author must provide list of Abbreviations that all are used in manuscript for readers. It is difficult to follow short forms in manuscript if full list of Abbreviations is not available.

A list of Abbreviation has been included at the beginning of the text before introduction, even though the number and utilization of those abbreviations might not justify such a list from our point of view:

“Abbreviations

ANLS: astrocyte-neuron lactate shuttle; ATP: Adenosine triphosphate; CVA: Cerebrovascular accident; OPCs: oligodendrocyte progenitor cells; TMS: transcranial magnetic stimulation; WHO: World Health Organization.”

  1. On page 4, Author state about that fundamental insight of Ramon y Cajal in 1899: All of the various conformations of the neuron and its various components are simply mor-.. Missing reference of Ramon y Cajal in 1899 in reference list so, please add that reference in final list.

The reference has been included:

  1. Ramon y Cajal, S. Histology of the Nervous System of Man and Vertebrates. Oxford Univ. Press; Oxford, UK, 1995.

  1. On page 5, The first three mechanisms concern within-network compensation whereas the last two refer to across-network compensation: Author already give no from first to fifth, so I would like to put it as in sentence that “The first three mechanisms concern within-network compensation whereas the last two 4th and 5th one refer to across- network compensation”

The sentence has been rephrased accordingly: “The first, second and third mechanisms concern ­within-network compensation whereas the fourth and fifth refer to across-network compensation:”

  1. No of subtitles are wrong after no 5. Interhemispheric dialog: opening and closing gates. Please correct and start that from onwards 6. Stroke and adaptative neuroplasticity: a complex “pas-de-deux”

Corrections have been made.

Reviewer 2 Report

Dear Editor,

The manuscript by Viruega and Gaviria reviews key issues of ipsilesional and contralesional hemispheric functioning allowing the foundations of effective neurorehabilitation strategies.

The review is comprehensive, informative and up-to-date (in most parts). Authors were successful in providing some well compiled opinions and summaries which will be of interest for Life readers and beyond.

However, there is a number of major and minor points that would need to be addressed in order to improve the quality of this paper before it can be accepted for publication:

General:

-It was to be way easier if the review has lines number.

 Major:

-Introduction “the extent of recovery is limited, and novel therapeutic approaches are urgently needed”. Neurodegenerative diseases are yet incurable conditions. Authors need to refer to recent advances including the use of high-throughput screening and computer-aided drug design as have been nicely reviewed by Aldewachi et al 2021 and Salman et al 2021 as they can provide a novel insight that can support the target validation. References to be included:

https://pubmed.ncbi.nlm.nih.gov/33672148/

https://pubmed.ncbi.nlm.nih.gov/33925236/

-Section 2 “Those include the transient role of contralateral hemisphere in recovery of lateralized functions, the disparate recovery levels of highly lateralized functions such language or motor function, and the reduced recovery of functions that are not typically highly lateralized [13]”. Author are encouraged to compare the findings from the above mention Sylvain et al study where they have shown regional effects.

-Section 3: This section will benefit from a general introduction about the energetic brain such as that the brain consumes 20% of total body energy despite its relatively small volume.

Moreover “Active maintenance of electrochemical gradients across neuronal mem- branes accounts for most of the substantial brain’s metabolic cost [23]; these are pulled down by myelination and pulled up by axonal length and diameter, long-distance con- nections being metabolically more expensive to maintain [25]. Besides, by minimizing the length of anatomical connections in the network (i.e., pulling down the wiring costs), the system will also regulate running-dependent costs”. This is rather a simplistic view. Authors need to mention the astrocyte‐neuron lactate shuttle (ANLS) hypothesis postulated in 1994 (Pellerin and Magistretti 1994). According to this, astrocytes serve as a ‘lactate source’ whereas neurons serve as a ‘lactate sink’. Moreover, the opposition by Bak and colleagues who argued that oxidative metabolism of lactate within neurons only occurs during repolarization (and in the period between depolarizations) rather than during neurotransmission activity. The emerging role of astrocytes has helped in settling this debate in favour for ANLS hypothesis. References to be included:

https://pubmed.ncbi.nlm.nih.gov/31318452/

https://pubmed.ncbi.nlm.nih.gov/19393013/

https://pubmed.ncbi.nlm.nih.gov/7938003/

-Section 5: the review lacks a detailed mention regarding the essential role of glial cells in neurological diseases. Glial cells, particularly astrocytes, appear to play critical and interactive roles especially at the BBB and BSCB. A recent work by Kitchen et al Cell 2020, has showed that targeting astrocytes is a viable therapeutic target. This role has been recently been confirmed in stroke by the work of Sylvain et al BBA 2021 where they have also shown a link to brain energy metabolism as indicated by the increase of glycogen levels.

Also, authors need to mention the role of astrocytes in the formation of tripartite synapse as established by Araque et al. since this has provided the foundation for the role of astrocytes in various CNS disorders. References:

https://pubmed.ncbi.nlm.nih.gov/10322493/

https://pubmed.ncbi.nlm.nih.gov/19615761/

 Minor:

-Section 6 “Homeostasis is a major issue for every post-injury situation, no matter how far away the injury has occurred”. Author are encouraged to discuss physiological measures such as the use of therapeutic hypothermia. References:

https://pubmed.ncbi.nlm.nih.gov/17127332/

https://www.ncbi.nlm.nih.gov/pmc/articles/PMC2953728/

Best

Author Response

Dear Reviewer,

Thank you very much for your comments that led us to improve the quality of our work.

You can find bellow the answer point-by-point of your remarks:

General:
It was to be way easier if the review has lines number.

Unfortunately, the template provided by the review doesn’t include lines number.

Major:

  1. Introduction “the extent of recovery is limited, and novel therapeutic approaches are urgently needed”. Neurodegenerative diseases are yet incurable conditions. Authors need to refer to recent advances including the use of high-throughput screening and computer-aided drug design as have been nicely reviewed by Aldewachi et al 2021 and Salman et al 2021 as they can provide a novel insight that can support the target validation. References to be included:

https://pubmed.ncbi.nlm.nih.gov/33672148/ https://pubmed.ncbi.nlm.nih.gov/33925236/

We have added a paragraph to include the recent advances in neurotherapeutics research (and the corresponding references), even though our focus was mainly the « beyond the pill » neurorehabilitation relying on the intrinsic characteristics of the nervous system (compensatory mechanisms):

“Many hopes and expectations have been recently placed on neuro-pharmacotherapeutics through the use of high-throughput screening and computer-aided drug design for an optimal identification and validation of molecular targets. However, even though potentially complementary to neurorehabilitation, neuropharmacology must deal with substantial barriers. Indeed, drug discovery and development are seriously hindered by the incomplete understanding of the pathophysiology of neurological diseases and most of the pharmacological treatments target the symptoms instead of the cause of the disease [10-11].”

  1. Aldewachi, H.; Al-Zidan, R.N.; Conner, MT.; Salman, M.M. High-Throughput Screening Platforms in the Discovery of Novel Drugs for Neurodegenerative Diseases. Bioengineering (Basel). 2021, 8(2), 30. Doi: 10.3390/bioengineering8020030.
  2. Salman, M.M.; Al-Obaidi, Z.; Kitchen, P.; Loreto, A.; Bill, R.M.; Wade-Martins, R. Advances in Applying Computer-Aided Drug Design for Neurodegenerative Diseases. Int. J. Mol. Sci. 2021, 22(9), 4688. Doi : 10.3390/ijms22094688.

  1. Section 2 “Those include the transient role of contralateral hemisphere in recovery of lateralized functions, the disparate recovery levels of highly lateralized functions such language or motor function, and the reduced recovery of functions that are not typically highly lateralized [13]”. Author are encouraged to compare the findings from the above mention Sylvain et al study where they have shown regional effects.

In the study of Sylvain, regional effects are indeed described with respect to the energetics of the injured site, where rise of glycogen is temporarily neuroprotective to at-risk cells in the penumbra during the acute post-stroke phase. Those molecular mechanisms of energy supply driven by astrocytes (glycogen, lactate) are mandatory for the preservation of intra-hemispheric neuroplasticity since they protect the cells of the penumbra. We have included a comment on section 3 instead of section 2, which was a better place for commenting this energy-dependent issue:

“Some intrinsic mechanisms of neuroprotection are triggered by the nervous system after an ischemic injury. For example, according to some studies in animal models of cerebrovascular ischemia, glycogen increases in the penumbra within three days post-injury and colocalizes with astrocytes [37]. The rise of glycogen might temporarily neuroprotect at-risk cells through increasing the available metabolic substrate. These molecular mechanisms of energy supply driven principally by astrocytes, although mostly insufficient for the overall tissue viability and function preservation, are mandatory for supporting intra-hemispheric neuroplasticity. They protect the cells of the penumbra that are essential contributors for behavioral restitution.”

  1. Sylvain, N.J.; Salman, M.M.; Pushie, M.J.; Hou, H.; Meher, V.; Herlo, R.; Peeling, L.; Kelly, M.E. The effects of trifluoperazine on brain edema, aquaporin-4 expression and metabolic markers during the acute phase of stroke using photothrombotic mouse model. Biochim. Biophys. Acta Biomembr. 2021, 1863(5), 183573. Doi: 10.1016/j.bbamem.2021.183573.

  1. Section 3: This section will benefit from a general introduction about the energetic brain such as that the brain consumes 20% of total body energy despite its relatively small volume.

A short general introduction about energetics of brain functioning has been added at the beginning of the section accordingly:

“The adult human brain is less than 2% of the body’s volume (~1.4 kg for 70 kg) but burns ~23% of daily caloric intake (i.e., ~415 kcal/day). In the newborn (~0.4 kg for 3.5 kg), this ratio is even more disproportionated since the brain burns around 75% of total daily caloric intake (i.e., ~120 kcal/day). Each neuron consumes approximately 4.8 x 10−6 cal/day: it may fire around 350,000 times/day at a rate ranging from 0.15 to 16 Hz and each action potential may consume around 1.19 x 108 ATP. One action potential of a cortical neuron per second raises oxygen consumption by 145 mL/100g gray matter/h [22]. The spontaneous brain activity accounts for 70% of the energy consumed by the brain and thus, at a whole brain level, basal metabolism is estimated to consume 30% of brain glucose. Thus, the high energetic cost of the human brain function can only be held through a combination of strategies for efficient energy use [23].”

  1. Burroni, J.; Taylor, P.; Corey, C.; Vachnadze, T.; Siegelmann, H.T. Energetic Constraints Produce Self-sustained Oscillatory Dynamics in Neuronal Networks. Front. Neurosci. 2017, 11, 80. Doi: 10.3389/fnins.2017.00080.
  2. Tomasi, D.; Wang, G.J.; Volkow, N.D. Energetic cost of brain functional connectivity. Proc. Natl Acad. Sci. USA. 2013, 110(33), 13642-13647. Doi: 10.1073/pnas.1303346110.

Moreover “Active maintenance of electrochemical gradients across neuronal mem- branes accounts for most of the substantial brain’s metabolic cost [23]; these are pulled down by myelination and pulled up by axonal length and diameter, long-distance con- nections being metabolically more expensive to maintain [25]. Besides, by minimizing the length of anatomical connections in the network (i.e., pulling down the wiring costs), the system will also regulate running-dependent costs”. This is rather a simplistic view. Authors need to mention the astrocyte‐neuron lactate shuttle (ANLS) hypothesis postulated in 1994 (Pellerin and Magistretti 1994). According to this, astrocytes serve as a ‘lactate source’ whereas neurons serve as a ‘lactate sink’. Moreover, the opposition by Bak and colleagues who argued that oxidative metabolism of lactate within neurons only occurs during repolarization (and in the period between depolarizations) rather than during neurotransmission activity. The emerging role of astrocytes has helped in settling this debate in favour for ANLS hypothesis. References to be included:

https://pubmed.ncbi.nlm.nih.gov/31318452/

https://pubmed.ncbi.nlm.nih.gov/19393013/  https://pubmed.ncbi.nlm.nih.gov/7938003/  

A comment on the participation of astrocytes to the energetic machinery of the nervous system (ANLS hypothesis) has been included on section 3. The three suggested references have also been included:

“Under normal physiologic conditions, interactions between neurons and glial cells are vital to meet the energy needs of the brain but are also important in the control of many essential brain functions such as homeostasis of the body and memory consolidation. Neuronal metabolic processes directly depend on the activity of astrocytes, which produce lactate and activate glycolysis and glycogen metabolism [31]. Indeed, astrocytes store glucose as glycogen, which can be temporarily used for oxidative metabolism, leading to the generation of lactate, which can, in turn, be shuttled to neurons as an energy source [32-35]. All these metabolic issues gave rise to the astrocyte-neuron lactate shuttle (ANLS) hypothesis, whereby glutamate released in the synapsis and its reuptake into astrocytes triggers glucose uptake into the brain parenchyma and lactate production by astrocytes for the use of neurons [34-35]. The transfer of lactate from astrocytes to neurons is one example of the wide palette of metabolic relationships between these cells. Lactate in the brain has long been associated with ischemia but it is now considered a main regulator of the brain’s ‘homeostatic tone’, by ensuring adequate energy supply, setting neuronal excitability levels, and regulating adaptive functions that are mediated by plasticity mechanisms (e.g., memory). The ANLS model has also been extended to metabolic exchanges between oligodendrocytes and axons showing that, in animal models, lactate released by oligodendrocytes is required to maintain axonal function [34].”

  1. Falkowska, A.; Gutowska, I.; Goschorska, M.; Nowacki, P.; Chlubek, D.; Baranowska-Bosiacka, I. Energy Metabolism of the Brain, Including the Cooperation between Astrocytes and Neurons, Especially in the Context of Glycogen Metabolism. Int. J. Mol. Sci. 2015, 16(11), 25959-25981. Doi: 10.3390/ijms161125939.
  2. Bak, L.K.; Walls, A.B.; Schousboe, A.; Ring, A.; Sonnewald, U.; Waagepetersen, H.S. Neuronal glucose but not lactate utilization is positively correlated with NMDA-induced neurotransmission and fluctuations in cytosolic Ca2+ levels. J. Neurochem. 2009, 109 Suppl 1, 87-93. Doi: 10.1111/j.1471-4159.2009.05943.x.
  3. Bordone, M.P.; Salman, M.M.; Titus, H.E.; Amini, E.; Andersen, J.V.; Chakraborti, B.; Diuba, A.V.; Dubouskaya, T.G.; Ehrke, E.; Espindola de Freitas, A.; et al. The energetic brain – A review from students to students. J. Neurochem. 2019, 151(2), 139-165. Doi: 10.1111/jnc.14829.
  4. Magistretti, P.J.; Allaman, I. Lactate in the brain: from metabolic end-product to signalling molecule. Nat. Rev. Neurosci. 2018, 19(4), 235-249. Doi: 10.1038/nrn.2018.19.
  5. Pellerin, L.; Magistretti, P.J. Glutamate uptake into astrocytes stimulates aerobic glycolysis: a mechanism coupling neuronal activity to glucose utilization. Proc. Natl Acad. Sci. USA. 1994, 91(22), 10625-10629. Doi: 10.1073/pnas.91.22.10625.

  1. Section 5: the review lacks a detailed mention regarding the essential role of glial cells in neurological diseases. Glial cells, particularly astrocytes, appear to play critical and interactive roles especially at the BBB and BSCB. A recent work by Kitchen et al Cell 2020, has showed that targeting astrocytes is a viable therapeutic target. This role has been recently been confirmed in stroke by the work of Sylvain et al BBA 2021 where they have also shown a link to brain energy metabolism as indicated by the increase of glycogen levels.

Also, authors need to mention the role of astrocytes in the formation of tripartite synapse as established by Araque et al. since this has provided the foundation for the role of astrocytes in various CNS disorders. References:

https://pubmed.ncbi.nlm.nih.gov/10322493/  https://pubmed.ncbi.nlm.nih.gov/19615761/  

A description of the role of glial cells in brain functioning has been added in section 6 as well as the two suggested references concerning the participation of astrocytes in the tripartite synapsis:

“In the last two decades, new evidence has been provided with respect to the mor-pho-functional characteristics of brain circuits and the role of glial cells in all aspects of brain function including neural plasticity. Radial glia, astrocytes, OPCs, oligodendrocytes and microglia each influence nervous system development (i.e., neuronal differentiation, migration, axon specification and growth, circuit assembly and synaptogenesis). With neural circuit maturation, each glial type fulfils key roles in synaptic communication, plasticity, homeostasis, and network-level activity [71]. Interestingly enough, astrocytes are intimately associated in the so-called “tripartite synapse”, receiving signals from the presynaptic neuron and responding by releasing feedback signals. These glial cells are integral modulatory components of the synapse. Besides fast neurotransmission, astrocyte regulation of synaptic transmission can run on a different time scale. Thus, astrocytes can transitorily control the synaptic strength, contributing to long-term synaptic plasticity. They are cellular processors of synaptic information and can regulate synaptic transmission (and plasticity) by treatment, transfer, and storage of information. Concomitantly to neuronal populations, they must be considered as key elements involved in brain function and potential targets for functional refinement in health and disease [72-73].”

  1. Allen, N.J.; Lyons, D.A. Glia as architects of central nervous system formation and function. Science. 2018, 362(6411), 181-185. Doi: 10.1126/science.aat0473.
  2. Araque, A.; Parpura, V.; Sanzgiri, R.P.; Haydon, P.G. Tripartite synapses: glia, the unacknowledged partner. Trends Neurosci. 1999, 22(5), 208-215. Doi: 10.1016/s0166-2236(98)01349-6.
  3. Perea, G.; Navarrete, M.; Araque, A. Tripartite synapses: astrocytes process and control synaptic information. Trends Neurosci. 2009, 32(8), 421-431. Doi: 10.1016/j.tins.2009.05.001.

The implementation of neuroprotective and/or neuroregenerative therapeutic strategies after stroke is beyond the scope of the current review. Nevertheless, we added a comment on the role of astrocytes in the early preservation of the at-risk cells (the reference of Sylvain et al. 2021 has also been added): see Point 2

Minor:

  1. Section 6 “Homeostasis is a major issue for every post-injury situation, no matter how far away the injury has occurred”. Author are encouraged to discuss physiological measures such as the use of therapeutic hypothermia. References:

https://pubmed.ncbi.nlm.nih.gov/17127332/  https://www.ncbi.nlm.nih.gov/pmc/articles/PMC2953728/  

We added a comment on physiological measures in section 7 accordingly:

“A myriad of theories regarding brain functioning after injury have been put forth but current clinical management of severe disabilities after CVA is often ineffective or not totally conclusive (e.g., the use of therapeutic hypothermia has been proposed as promising, but results are partly questionable mainly due to its efficacy in hemorrhagic stroke, its side effects, and the invasive nature of the procedure [84]).”

  1. Yenari, M.A.; Hemmen, T.M. Therapeutic hypothermia for brain ischemia: where have we come and where do we go? Stroke. 2010, 41(10 Suppl), S72-S74. Doi: 10.1161/STROKEAHA.110.595371.

Round 2

Reviewer 2 Report

Dear Editor,

The authors have successfully addressed the majority of my comments and concerns in order to improve the quality of the manuscript.

I do believe that the corrections, additional sections and updated references, have contributed to enhancing the clarity of the manuscript, which I can now endorse for publication.

All the best!